# Beyond Euclidean Clipping: Overcoming Exploration Collapse in LLM RL via Riemannian Isometric Policy Optimization

Zhicheng Cai [1 2 3]  Xinyuan Guo [1 3]  Hanlin Wu [1 2 3]  Mingxuan Wang [2 3]  Wei-Ying Ma [1 3]  Ya-Qin Zhang [1 3]
Hao Zhou [1 3] *

## Abstract

Reinforcement learning (RL) has become a dominant paradigm for enhancing LLMs' reasoning capabilities. However, RL algorithms with PPO-Clip are inherently limited by exploration collapse. Subsequent works remain primarily heuristic and fail to identify the essential cause of PPO-Clip's failure. This work reveals the fundamental flaw of PPO-Clip: it implicitly measures policy discrepancy using Euclidean metric, which is theoretically inconsistent with the intrinsic geometry on the policy Riemannian manifold. This geometric mismatch results in overly conservative updates in low-probability regions while aggressive in high-probability regions, ultimately collapsing exploration. To correct this geometric flaw, we propose Riemannian Isometric Policy Optimization (RIPO), which guarantees isometric policy updates on the Riemannian manifold, effectively balancing exploration and exploitation. We further show that RIPO achieves a favorable bias-variance trade-off, which stabilizes optimization. Extensive experiments demonstrate that RIPO significantly surpasses existing LLM RL algorithms across seven competition-level benchmarks (up to 60% improvement over GRPO on AIME24).

## 1. Introduction

Reinforcement learning (RL) has emerged as a central paradigm for enhancing the reasoning capabilities of large language models (LLMs) (Guo et al., 2025; Comanici et al., 2025; OpenAI, 2024). It has demonstrated remarkable success in domains requiring long-horizon decision-making, such as mathematical reasoning (Shao et al., 2024; Wang et al., 2024; Luo et al., 2025), and so on (OpenAI, 2025; Jin et al., 2025; Dou et al., 2024; Anthropic, 2024).

Despite these advances, the PPO-Clip (Schulman et al., 2017) used in modern LLM RL algorithms (Guo et al., 2025) incurs a critical issue: *exploration collapse* (Yu et al., 2025). Specifically, PPO-Clip makes the policy rapidly concentrates on a narrow set of high-probability actions during training, thus rare but vital actions are unexplored, severely suppressing model performance. This is especially devastating in long-horizon reasoning tasks, where the state-action space is extremely vast, thus extensive exploration is indispensable to discover a feasible solution. Ultimately, exploration collapse hinders further scaling progress of RL, stifling the potential in increasingly challenging tasks.

Subsequent works (Yang et al., 2025b; Zheng et al., 2025) provide symptomatic fixes without identifying the root cause of PPO-Clip's failure. For example, DAPO (Yu et al., 2025) raises the clipping boundary to encourage exploration. While yielding empirical gains, they remain heuristic within the PPO-Clip framework, offering limited theoretical understanding and leaving exploration collapse unresolved.

In this work, we discover the fundamental flaw of PPO-Clip lies in the *geometric mismatch between Euclidean metric and Riemannian manifold*. Specifically, PPO-Clip implicitly measures policy discrepancy using a Euclidean metric on the importance ratio, treating equal ratio deviations as equal policy changes. However, the intrinsic discrepancy between policies is governed by the Kullback-Leibler (KL) divergence (Schulman et al., 2015), which induces a Riemannian geometry on the statistical manifold of policies. This geometric mismatch leads to pathological update behavior: updates for low-probability actions consume almost no KL budget thus become overly conservative, while updates for high-probability actions are disproportionately aggressive. Consequently, rare but informative actions are systematically under-updated, causing premature policy concentration and eventual exploration collapse.

To correct the mismatch, we propose *Riemannian Isometric Policy Optimization* (RIPO), a theoretically grounded RL algorithm that fundamentally reinterprets policy updates on

[1]Institute for AI Industry Research (AIR), Tsinghua University
[2]ByteDance Seed [3]SIA-Lab of Tsinghua AIR and ByteDance Seed. Correspondence to: Hao Zhou <zhouhao@air.tsinghua.edu.cn>.

*Proceedings of the 43rd International Conference on Machine Learning*, Seoul, South Korea. PMLR 306, 2026. Copyright 2026 by the author(s).

the Riemannian manifold. By dynamically adapting the clipping boundary to the local Riemannian geometry, RIPO permits larger updates for low-probability actions while constraining updates for high-probability actions. This *isometric policy update* balances exploration and exploitation, significantly improving the model performance, particularly in long-horizon tasks that require sustained exploration. In addition, such geometric isometry induces statistical homoscedasticity, leading to a favorable bias–variance trade-off and enabling more stable policy optimization.

In summary, our contributions are threefold:

- We identify a fundamental *geometric mismatch of PPO-Clip*, showing that its Euclidean clipping is incompatible with the intrinsic Riemannian manifold, leading to pathological updates and exploration collapse.

- We propose *Riemannian Isometric Policy Optimization* (RIPO), a theoretically grounded RL algorithm that enforces isometric updates on the Riemannian manifold, effectively balancing exploration and exploitation. We further show that RIPO induces a favorable bias–variance trade-off.

- We conduct extensive experiments across various LLMs and competition-level benchmarks. RIPO significantly outperforms existing RL algorithms, achieving up to 60% relative improvement over GRPO on AIME24, demonstrating its superiority and efficacy in enhancing reasoning performance.

## 2. Preliminary

### 2.1. Trust Region Policy Optimization

Trust Region Policy Optimization (TRPO) (Schulman et al., 2015) theoretically establishes that optimizing the surrogate objective under a KL-divergence constraint (the *trust region*) between the target policy $\pi_\theta$ and the behavior policy $\pi_{\theta_{\text{old}}}$ guarantees monotonic policy improvement. Specifically,

$$\max_\theta \quad \mathbb{E}_{s \sim \rho_{\theta_{\text{old}}}, a \sim \pi_{\theta_{\text{old}}}} \left[ \frac{\pi_\theta(a|s)}{\pi_{\theta_{\text{old}}}(a|s)} \hat{A}(s,a) \right]$$
$$s.t. \quad \mathbb{E}_{s \sim \rho_{\theta_{\text{old}}}} [D_{\text{KL}}(\pi_{\theta_{\text{old}}}(\cdot|s) || \pi_\theta(\cdot|s))] \le \delta \quad (1)$$

where $\hat{A}$ is the estimated advantage and $\delta$ is the trust-region radius. Despite its solid theoretical guarantees and appealing empirical performance, such constrained optimization incurs significant computational overhead, making TRPO difficult to scale in practice.

### 2.2. Proximal Policy Optimization and Clip

To alleviate the computational burden and retain the intuition of TRPO, Proximal Policy Optimization (PPO) (Schulman et al., 2017) heuristically approximates the second-order optimization with a first-order clipped surrogate objective. For clarity, we omit the explicit dependence on $(s, a)$:

$$\mathcal{J}_{\text{PPO}}(\theta) = \mathbb{E} \left[ \min \left( r(\theta) \hat{A}, \text{clip}(r(\theta), 1-\epsilon, 1+\epsilon) \hat{A} \right) \right]$$

where $r_{s,a}(\theta) = \frac{\pi_\theta(a|s)}{\pi_{\theta_{\text{old}}}(a|s)}$ is the importance ratio, $\epsilon$ is the fixed clipping boundary, and $\hat{A}$ is the advantage estimated with GAE (Schulman et al., 2016). By clipping the importance ratio, PPO constrains the policy updates within a proximal region, thus stabilizing training and improving policy performance. Due to the simplicity and effectiveness, clipping has become a central component in modern RL algorithms for large language models.

### 2.3. Group Relative Policy Optimization and Variants

Similar to PPO, Group Relative Policy Optimization (GRPO) (Guo et al., 2025; Shao et al., 2024) retains the PPO-Clip but eliminates the value model and estimates advantage in a group-relative manner for computational and memory efficiency. In the LLM RL setting, given a query $q \sim \mathcal{D}$, the behavior policy $\pi_{\theta_{\text{old}}}$ samples a group of $G$ individual responses $\{o_i\}_{i=1}^G$ with sparse rewards $\{R_i\}_{i=1}^G$. The GRPO objective is,

$$\mathcal{J}_{\text{GRPO}}(\theta) = \mathbb{E}_{q \sim \mathcal{D}, \{o_i\}_{i=1}^G \sim \pi_{\theta_{\text{old}}}(\cdot|q)} \left[ \frac{1}{G} \sum_{i=1}^G \frac{1}{|o_i|} \sum_{t=1}^{|o_i|} \right.$$
$$\left. \min \left( r_{i,t}(\theta) \hat{A}_{i,t}, \text{clip}(r_{i,t}(\theta), 1-\epsilon, 1+\epsilon) \hat{A}_{i,t} \right) \right] \quad (2)$$

where $r_{i,t}(\theta) = \frac{\pi_\theta(o_{i,t}|q, o_{i<t})}{\pi_{\theta_{\text{old}}}(o_{i,t}|q, o_{i<t})}$ is the importance ratio, and $\hat{A}_{i,t} = \frac{R_i - \text{mean}(\{R_i\}_{i=1}^G)}{\text{std}(\{R_i\}_{i=1}^G)}$ is the group-normalized advantage.

A series of GRPO variants have been proposed to mitigate two commonly observed issues in PPO-Clip: exploration collapse and gradient instability. DAPO (Yu et al., 2025) introduces decoupled Clip-Higher, setting the clipping range as $(1 - \epsilon_{\text{low}}, 1 + \epsilon_{\text{high}})$ and $\epsilon_{\text{high}} > \epsilon_{\text{low}}$ to encourage exploration. DCPO (Yang et al., 2025b) adapts clipping thresholds dynamically. GSPO (Zheng et al., 2025) and GMPO (Zhao et al., 2025) clip on the sequential importance ratio to reduce gradient variance. CISPO (Chen et al., 2025), GPPO (Su et al., 2025), and SAPO (Gao et al., 2025) propose to maintain the gradient of these clipped tokens.

Although these variants yield empirical gains, they remain largely heuristic with limited theoretical understanding. Importantly, they do not identify and address the fundamental flaw of PPO-Clip, which we examine next.

# 3. The Geometric Flaw of PPO Clip

In this section, we first revisit the exploration collapse phenomenon of PPO-Clip. Then we theoretically illustrate that such issue originates from the geometric mismatch between the Euclidean metric used by PPO-Clip and the intrinsic geometry of the Riemannian manifold induced by KL divergence.

## 3.1. Exploration Collapse of PPO-Clip

The essence of PPO-Clip is to act as a first-order approximation to the trust region by restricting the importance ratio $r_{s,a}(\theta) = \frac{\pi_\theta(a|s)}{\pi_{\theta_{\text{old}}}(a|s)}$ to the interval $(1 - \epsilon, 1 + \epsilon)$. However, previous work (Yu et al., 2025) shows that PPO-Clip suppresses policy exploration, making it much easier to further increase the probability of already-preferred "exploitation tokens" than to promote low-likelihood "exploration tokens", which ultimately leads to exploration collapse.

Specifically, given $\epsilon = 0.2$ (the default value of most cases), consider action with relatively high probability of $\pi_{\theta_{\text{old}}} = 0.8$, the maximum possible updated probability $\pi_\theta(a|s)$ is $0.96$ with an increase of $0.16$. While for low-probability action with $\pi_{\theta_{\text{old}}} = 0.01$, clipping limits the updated probability to only $0.012$, corresponding to a negligible increase of $0.002$. Consequently, even when low-probability while valuable actions are sampled, PPO-Clip fails to generate probability updates commensurate with the rarity. In contrast, high-frequent while low-value actions are more likely to receive large updates. As training proceeds, this asymmetry gradually reduces policy behavioral diversity, namely, the responses of policy tend to be nearly identical.

DAPO (Yu et al., 2025) proposes decoupled Clip-Higher to encourage exploration, setting the range as $(1 - \epsilon_{\text{low}}, 1 + \epsilon_{\text{high}})$ and lifting $\epsilon_{\text{high}}$ from $0.2$ to $0.28$. Consequently, the maximum update for low-probability action (*e.g.*, $0.01$) is lifted from $0.012$ to $0.0128$. However, the increase is merely $0.0008$, which is still negligible. Even more critically, the maximum update for high-probability action (*e.g.*, $0.8$) is also lifted from $0.96$ to $1$, which further intensifies the reduction in behavioral diversity. Therefore, DAPO fails to fundamentally analyze the flaw of PPO-Clip and resolve it.

In summary, PPO-style Clip indiscriminately updates or truncates tokens, regardless of whether the corresponding action is already highly exploited or rarely explored. We next illustrate that such issue fundamentally originates from the geometric mismatch.

## 3.2. Geometric Mismatch in Policy Divergence

**The Euclidean Assumption of PPO-Clip.** Equivalently, PPO-Clip constrains the ratio through $|r(\theta) - 1| < \epsilon$, which induces an implicit Euclidean distance measure:

$$d_{\text{clip}}(\pi_{\theta_{\text{old}}}, \pi_\theta) = (\frac{\pi_\theta}{\pi_{\theta_{\text{old}}}} - 1)^2 = (r(\theta) - 1)^2 \quad (3)$$

for quantifying the change between two policies. Under this view, as long as two pairs of samples have the **same ratio deviation**, PPO-Clip regards them as having the **same trust level**, *i.e.*, the **same discrepancy** between the new and old policy distributions, and therefore subjects them to the **same constraint**. This implicitly assumes that policy change is uniform over the entire statistical manifold of policies. That is, the Euclidean deviation of the ratio $d_{\text{clip}}$ can uniformly characterize the divergence between policy distributions. However, this assumption is fundamentally incorrect from a geometric perspective.

**Riemannian Geometry of Policy Discrepancy.** As stated in TRPO (Schulman et al., 2015), the discrepancy between two policies is measured by the KL divergence $D_{\text{KL}}(\pi_{\theta_{\text{old}}}(\cdot|s)||\pi_\theta(\cdot|s))$, which admits the following second-order Taylor expansion:

$$\begin{aligned}
\Phi(\theta) &= D_{\text{KL}}(\pi_{\theta_{\text{old}}}(\cdot|s)||\pi_\theta(\cdot|s)) \\
&\approx \left( \Phi(\theta) + \Delta\theta \nabla_\theta \Phi(\theta) + \frac{1}{2}\Delta\theta^\top \nabla_\theta^2 \Phi(\theta)\Delta\theta \right)\Bigg|_{\theta=\theta_{\text{old}}} \\
&= \frac{1}{2}\Delta\theta^\top F(\theta)|_{\theta=\theta_{\text{old}}} \Delta\theta
\end{aligned}$$

$$(4)$$

where $\Delta\theta = \theta - \theta_{\text{old}}$, and $F(\theta)$ is the Fisher Information Matrix, delivering a Riemannian manifold. To relate parameter updates to their effect on probability distributions, we map the above approximation from the parameter space $\Theta = \{\theta\}$ to the induced policy space of probability distributions $\Pi = \{\pi_\theta(a|s) \mid \theta \in \Theta\}$. Formally, we have:

$$\begin{aligned}
2D_{\text{KL}}&(\pi_{\theta_{\text{old}}}(\cdot|s)||\pi_\theta(\cdot|s)) \approx \Delta\theta^\top F(\theta)|_{\theta=\theta_{\text{old}}} \Delta\theta \\
&= \Delta\theta^\top \mathbb{E}_a \left[ \nabla_\theta \log \pi_\theta(a|s) \nabla_\theta \log \pi_\theta(a|s)^\top \right]\Bigg|_{\theta=\theta_{\text{old}}} \Delta\theta \\
&= \Delta\theta^\top \sum_a \pi_\theta(a|s) \frac{\nabla_\theta \pi_\theta(a|s)}{\pi_\theta(a|s)} \frac{\nabla_\theta \pi_\theta(a|s)^\top}{\pi_\theta(a|s)}\Bigg|_{\theta=\theta_{\text{old}}} \Delta\theta \\
&= \sum_a \frac{1}{\pi_{\theta_{\text{old}}}(a|s)} \left( \nabla_\theta \pi_\theta(a|s)^\top|_{\theta=\theta_{\text{old}}} \Delta\theta \right)^2
\end{aligned}$$

$$(5)$$

Perform first-order Taylor expansion of $\pi_\theta(a|s)$ around $\theta_{\text{old}}$:

$$\pi_\theta(a|s) \approx \pi_{\theta_{\text{old}}}(a|s) + \nabla_\theta \pi_\theta^\top(a|s)|_{\theta=\theta_{\text{old}}}\Delta\theta \quad (6)$$

Thus we obtain:

$$\begin{aligned}
D_{\text{KL}}(\pi_{\theta_{\text{old}}}(\cdot|s)||\pi_\theta(\cdot|s)) &\approx \frac{1}{2}\sum_a \frac{(\pi_\theta(a|s) - \pi_{\theta_{\text{old}}}(a|s))^2}{\pi_{\theta_{\text{old}}}(a|s)} \\
&= \frac{1}{2}\sum_a \pi_{\theta_{\text{old}}}(a|s)(r_{s,a}(\theta) - 1)^2
\end{aligned}$$

$$(7)$$

As a consequence, the geometric distance between two policies on the Riemannian manifold can be derived as

$$d_{\text{geom}}(\pi_{\theta_{\text{old}}}, \pi_\theta) \propto \pi_{\theta_{\text{old}}} \cdot (r(\theta) - 1)^2 \tag{8}$$

Crucially, this geometric distance depends on the underlying probability simplex $\pi_{\theta_{\text{old}}}$: distances in high-probability regions expand, while distances in low-probability regions shrink. Thus the geometry of Riemannian manifold is non-uniform. In contrast, PPO-Clip implicitly adopts a Euclidean metric that ignores the dependence on $\pi_{\theta_{\text{old}}}$, treating $(r(\theta) - 1)^2$ as globally equivalent, leading to a systematic mismatch in measuring policy divergence.

**Trust Region Perspective.** We now revisit the illustrative example of Sec. 3.1 in light of the derived geometry. Note that PPO-Clip intends to approximate a trust region constraint $D_{\text{KL}} \leq \delta$. For high-probability action $\pi_{\theta_{\text{old}}}(a|s) = 0.8$, consider $r(\theta) = 1.2$ which reaches the clipping boundary $|r(\theta) - 1| = 0.2$. Correspondingly, the maximum distance is $0.5 \times 0.8 \times 0.2^2 = 0.016$, which can be regarded as a reasonable trust region. While for low-probability action $\pi_{\theta_{old}}(a|s) = 0.01$, also consider $|r(\theta) - 1| = 0.2$. In this case, the induced distance is $0.5 \times 0.01 \times 0.2^2 = 0.0002$, which is orders of magnitude smaller than the available trust-region budget. Under PPO-Clip, both updates are treated identically due to their same ratio deviation. However, in Riemannian geometry, the second update moves almost zero distance, leaving substantial trust region budget unused. Consequently, high-probability tokens are allowed to grow aggressively, whereas low-probability tokens face a severely restricted trust region.

We conclude this section with following proposition.

**Proposition 3.1.** *PPO-Clip incorrectly employs a Euclidean metric to measure the discrepancy between policies, failing to align with the geometry of policy Riemannian manifold. This leads to overly conservative updates in low-probability regions while aggressive in high-probability regions, ultimately causing exploration collapse.*

# 4. Riemannian Isometric Policy Optimization

In this section, we first propose a theoretically grounded clipping mechanism, Riemannian Isometric Clip. We then illustrate that it achieves a desirable bias–variance trade-off. Finally, we introduce the complete Riemannian Isometric Policy Optimization algorithm.

## 4.1. Riemannian Isometric Clip

In order to address the inherent issue of PPO-Clip, we propose that updates for each state-action sample should move within the same geometric distance on the Riemannian manifold, rather than having the same Euclidean ratio deviation.

That is, each update should consume within the same budget of trust region. To achieve this, we propose Riemannian Isometric Clip (RIC) that dynamically adjusts the clipping boundary depends on the corresponding local probability simplex $\pi_{\theta_{\text{old}}}$. Formally, we require that the geometric distance for each update satisfies:

$$d_{\text{geom}}(\pi_{\theta_{\text{old}}}(a|s), \pi_\theta(a|s)) \triangleq \frac{1}{2}\pi_{\theta_{\text{old}}}(a|s)(r_{s,a}(\theta) - 1)^2 \leq \delta \tag{9}$$

Solving for $r_{s,a}(\theta)$, we obtain the theoretically grounded ratio constraint for each action:

$$|r_{s,a}(\theta) - 1| \leq \sqrt{\frac{2\delta}{\pi_{\theta_{\text{old}}}(a|s)}}$$

$$1 - \sqrt{\frac{2\delta}{\pi_{\theta_{\text{old}}}(a|s)}} \leq r_{s,a}(\theta) \leq 1 + \sqrt{\frac{2\delta}{\pi_{\theta_{\text{old}}}(a|s)}} \tag{10}$$

This result provides a distribution-dependent dynamic clipping threshold, which can be practically implemented as:

$$|r_{s,a}(\theta) - 1| \leq \epsilon_{s,a}(\pi_{\theta_{\text{old}}}), \quad \epsilon_{s,a}(\pi_{\theta_{\text{old}}}) = \sqrt{\frac{\delta}{\pi_{\theta_{\text{old}}}(a|s)}} \tag{11}$$

where the coefficient 2 is absorbed into the hyper-parameter $\delta$, which indicates the maximum geometric distance (*i.e.*, the radius of the trust region). Consequently, by considering the local probability simplex, RIC allows low-probability actions to receive larger policy updates, while high-probability actions are assigned reduced updates, thus maintaining a balanced exploration-exploitation trade-off.

**Revisit the Illustrative Example.** We again revisit the example of Sec. 3.1 to intuitively demonstrate the effect of RIC. With $\delta = 0.02$, for a high-probability action $\pi_{\theta_{\text{old}}}(a|s) = 0.8$, the maximum updated probability is constrained to $0.92$, which is noticeably less aggressive than $0.96$ under PPO-Clip. In contrast, for a low-probability action $\pi_{\theta_{\text{old}}}(a|s) = 0.01$, RIC allows the probability to increase to $0.024$, a substantially larger update compared with $0.012$ under PPO-Clip. Moreover, the consumed trust regions (geometric distances) of these two actions are constantly $0.01$ under RIC, compared to highly different values of $0.016$ and $0.0002$ under PPO-Clip. Consequently, RIC enables exploration without amplifying already dominant actions through allocating isometric trust regions.

## 4.2. Geometric Isometry Implies Homoscedasticity

We next analyze Riemannian Isometric Clip from the perspective of the critical *bias–variance trade-off* in off-policy RL algorithms (Schulman et al., 2015; Yang et al., 2025b), and reveal a deep connection between geometric isometry and statistical homoscedasticity.

For off-policy algorithms, the objective under target policy $\pi_\theta$ is estimated by samples $x$ (*i.e.*, state-action pairs) from behavior policy $\pi_{\theta_{\text{old}}}$ through importance sampling, namely,

$$\mathbb{E}_{x \sim \pi_\theta}[A(x)] = \mathbb{E}_{x \sim \pi_{\theta_{\text{old}}}}[r(x)A(x)], \quad r(x) = \frac{\pi_\theta(x)}{\pi_{\theta_{\text{old}}}(x)} \tag{12}$$

which is unbiased but often suffers from severe variance. The variance is dominated by the second-moment term while the squared-mean term is typically negligible (Tokdar & Kass, 2010; Murphy, 2012):

$$\mathbb{V}_{x \sim \pi_{\theta_{\text{old}}}}[r(x)A(x)] \approx \mathbb{E}_{x \sim \pi_{\theta_{\text{old}}}}\left[r(x)^2 A(x)^2\right]$$
$$= \sum_x \pi_{\theta_{\text{old}}}(x)\, r(x)^2 A(x)^2 \tag{13}$$

The variance explosion primarily originates from the long-tail distribution of $\pi_{\theta_{\text{old}}}(x)$. Define the variance contribution of sample $x$ is $v(x) = \pi_{\theta_{\text{old}}}(x)r(x)^2$, when $\pi_{\theta_{\text{old}}}(x)$ is small, $v(x) = \pi_\theta(x)^2/\pi_{\theta_{\text{old}}}(x)$ becomes arbitrarily large, causing uncontrollable contribution to the variance.

PPO-Clip implicitly mitigates this issue by truncating the importance ratio, $r(x) \leq 1+\epsilon$, thus discarding samples with large ratios. For samples $x'$ around the clipped region, the variance contribution is $v(x') = \pi_{\theta_{\text{old}}}(x')(1+\epsilon)^2 \to 0$ as $\pi_{\theta_{\text{old}}}(x') \to 0$, effectively reducing variance. However, PPO-Clip introduces severe bias by ignoring the contribution of these clipped samples to the objective.

In contrast, RIC adopts a distribution-dependent clipping threshold $r(x) \leq 1 + \sqrt{\delta/\pi_{\theta_{\text{old}}}(x)}$. Consequently, the variance contribution $v(x')$ of each sample $x'$ around the clipped region satisfies,

$$v(x') = \pi_{\theta_{\text{old}}}(x')\left(1 + \sqrt{\frac{\delta}{\pi_{\theta_{\text{old}}}(x')}}\right)^2 \approx \mathcal{O}(\delta) \tag{14}$$

yielding a *density-independent and constant-order* variance. As a result, RIC achieves a principled *bias–variance trade-off*: the variance is strictly smaller than standard importance sampling, while the bias is substantially smaller than PPO-Clip as more rare samples are considered.

**Geometric Interpretation.** This analysis reveals a fundamental connection between geometry and statistics: by guaranteeing isometric updates on the policy Riemannian manifold, RIC equalizes the second-order contribution of each sample to the trust region, which in turn induces statistical homoscedasticity in importance sampling. In contrast, PPO-Clip implicitly assumes a Euclidean geometry, leading to heteroscedastic variance and overly conservative updates in low-probability regions.

We conclude the proposed clipping mechanism as,

**Proposition 4.1.** *Riemannian Isometric Clip guarantees equal geometric distance for each state-action update on the policy Riemannian manifold, where low-probability actions are allowed larger updates and high-probability actions receive less aggressive updates. This geometrically principled clipping mechanism simultaneously mitigates exploration collapse and balances the bias–variance trade-off in policy optimization.*

### 4.3. Riemannian Isometric Policy Optimization

Now we present the objective of Riemannian Isometric Policy Optimization (RIPO [1]):

$$\mathcal{J}_{\text{RIPO}}(\theta) = \mathbb{E}_{q \sim \mathcal{D}, \{o_i\}_{i=1}^G \sim \pi_{\theta_{\text{old}}}(\cdot|q)}\left[\frac{1}{\sum_{i=1}^G |o_i|} \sum_{i=1}^G \sum_{t=1}^{|o_i|} \min\right.$$

$$\left. \left(r_{i,t}(\theta)\hat{A}_{i,t}, \text{clip}\left(r_{i,t}(\theta), 1-\epsilon_{i,t}(\pi_{\theta_{\text{old}}}), 1+\epsilon_{i,t}(\pi_{\theta_{\text{old}}})\right)\hat{A}_{i,t}\right)\right] \tag{15}$$

where $\epsilon_{i,t}(\pi_{\theta_{\text{old}}})$ is the dynamic clipping boundary of RIC as proposed in Eqn. 11. RIPO is proposed for LLM RL tasks and adopts the group-relative advantage estimator of GRPO. In addition, RIPO adopts token-level policy gradient loss as DAPO (Yu et al., 2025) to balance the gradient influence of both long and short trajectories.

## 5. Experiments

To validate the effectiveness of RIPO, we conduct extensive experiments on four LLMs of different sizes and types, across seven challenging benchmarks, and comparing with six representative RL algorithms.

### 5.1. Experimental Setup

**Models and Baselines.** We evaluate the RL algorithm's performance on four large language models of different scales and architectures, i.e., Llama3.2-3B-Instruct (Dubey et al., 2024), Qwen3-1.7B-Base, Qwen3-4B-Base, and Qwen3-8B-Base (Yang et al., 2025a).

We compare the proposed RIPO to six representative RL algorithms with different clipping mechanisms, including GRPO (Shao et al., 2024) with PPO-Clip, DAPO (Yu et al., 2025) with Clip-Higher, GSPO (Zheng et al., 2025) with sequence-level clipping, GMPO (Zhao et al., 2025) with geometric-mean ratio clipping, and DCPO (Yang et al., 2025b) with dynamic-adaptive clipping, following their default hyper-parameter settings. We set the $\delta$ of RIPO as $0.05$ by default. As a common practice (Yang et al., 2025b; Ye et al., 2020), we employ dual clipping to the importance ratio of GRPO, DAPO, DCPO, and RIPO, with the lower

---

[1]RIPO, pronounced like "ripple", highlighting how policy improvements propagate smoothly across the Riemannian manifold.

*Table 1.* Comparison of different RL algorithms (evaluated by Avg@8) across various mathematical-reasoning benchmarks.

| Method | AIME24 | AIME25 | AMC23 | HMMT25 | BRUMO25 | CMIMC25 | SMT25 | Average |
|--------|--------|--------|-------|--------|---------|---------|-------|---------|
| *Qwen3-1.7B-Base* | | | | | | | | |
| Base | 2.1 | 0.0 | 28.4 | 0.0 | 11.3 | 0.0 | 6.6 | 6.9 (-38.4%) |
| GRPO | 11.3 | 10.8 | 33.8 | 0.0 | 15.0 | 0.3 | 7.3 | 11.2 (+ 0.0%) |
| DAPO | 15.0 | 12.1 | 39.7 | 0.8 | 9.2 | 1.9 | 7.8 | 12.4 (+10.7%) |
| GSPO | 16.3 | 11.2 | 40.0 | **1.6** | 12.9 | 2.8 | 8.5 | 13.3 (+19.0%) |
| GMPO | 17.5 | 12.5 | 40.9 | 0.8 | 15.0 | **3.8** | 6.6 | 13.9 (+24.1%) |
| DCPO | 17.1 | 11.3 | **46.9** | 0.8 | **15.4** | 1.3 | **10.8** | 14.8 (+32.1%) |
| **RIPO** | **18.3** | **12.9** | 46.1 | 1.3 | **15.4** | 3.2 | 10.4 | **15.4 (+37.2%)** |
| *Llama3.2-3B-Instruct* | | | | | | | | |
| Base | 5.4 | 0.0 | 17.8 | 0.0 | 1.3 | 0.0 | 0.0 | 3.5 (-45.3%) |
| GRPO | 16.3 | 1.3 | 24.5 | 0.4 | 1.7 | 0.0 | 0.7 | 6.4 (+ 0.0%) |
| DAPO | 20.1 | 1.7 | 25.3 | **0.8** | 2.5 | 0.3 | 0.9 | 7.4 (+15.6%) |
| GSPO | 17.5 | 1.7 | 25.7 | 0.4 | 3.8 | 0.6 | 0.9 | 7.2 (+12.5%) |
| GMPO | 16.7 | 1.3 | 27.7 | 0.4 | 2.8 | **0.9** | 1.7 | 7.4 (+15.6%) |
| DCPO | 16.7 | **2.1** | 24.7 | 0.4 | 2.5 | 0.6 | 1.4 | 6.9 (+ 7.8%) |
| **RIPO** | **22.1** | **2.1** | **28.1** | **0.8** | **4.3** | **0.9** | **1.7** | **8.6 (+34.4%)** |
| *Qwen3-4B-Base* | | | | | | | | |
| Base | 10.0 | 8.3 | 50.0 | 0.0 | 18.3 | 1.3 | 8.9 | 13.8 (-46.3%) |
| GRPO | 30.4 | 20.8 | 63.4 | 7.5 | 25.8 | 12.4 | 19.8 | 25.7 (+ 0.0%) |
| DAPO | 31.7 | 21.3 | 69.4 | 9.2 | 26.3 | 10.0 | 21.7 | 27.1 (+ 5.4%) |
| GSPO | 32.1 | 21.3 | 70.3 | 9.2 | 26.7 | 10.0 | 18.2 | 26.8 (+ 4.3%) |
| GMPO | 32.5 | 26.7 | 68.8 | **11.3** | 26.3 | 10.6 | **22.5** | 28.4 (+10.5%) |
| DCPO | 31.7 | 24.2 | 71.3 | 8.8 | 25.0 | **15.0** | 19.8 | 27.8 (+ 8.2%) |
| **RIPO** | **32.9** | **27.5** | **73.1** | 10.0 | **31.3** | 14.7 | 21.5 | **30.1 (+17.1%)** |
| *Qwen3-8B-Base* | | | | | | | | |
| Base | 3.8 | 3.3 | 43.4 | 1.3 | 15.0 | 3.1 | 10.8 | 11.5 (-59.6%) |
| GRPO | 31.7 | 20.8 | 66.6 | 12.9 | 33.3 | 12.5 | 21.9 | 28.5 (+ 0.0%) |
| DAPO | 33.7 | 22.1 | 70.9 | 7.9 | 33.3 | **18.1** | 28.3 | 30.6 (+ 7.4%) |
| GSPO | 37.5 | 19.6 | 70.1 | 15.0 | 36.3 | 13.8 | 32.5 | 32.1 (+12.6%) |
| GMPO | 41.7 | **30.0** | 76.9 | 11.7 | 35.8 | 17.8 | 33.0 | 35.3 (+23.9%) |
| DCPO | 36.3 | 27.5 | 72.2 | 15.4 | 42.8 | 16.3 | 31.6 | 34.5 (+21.1%) |
| **RIPO** | **43.8** | 29.2 | **79.7** | **16.7** | **47.5** | **18.1** | **34.2** | **38.5 (+35.1%)** |

and upper bounds set to $0.5$ and $10$, respectively. We also remove the KL penalty term for all the methods following prior works (Yu et al., 2025; Yang et al., 2025b). Note that all these seven RL algorithms utilize the group-relative advantage estimator.

**Training.** We focus on evaluating the improvement in mathematical reasoning capability induced by RL algorithms, which is particularly challenging and widely regarded as a strong indicator of general reasoning capability. We adopt DAPO-Math-17k (Yu et al., 2025) as the training set, which includes 17,917 questions. For each question, we generate 8 rollouts and set the maximum response length as 16,384 tokens for thorough reasoning. In each RL iteration, the behavior policy $\pi_{\theta_{old}}$ totally generates 1024 rollouts with a train batch size of 128, and the current policy $\pi_\theta$ is updated 8 times with a mini-batch size of 16. Mathematical problems admit verifiable rewards, thus the rewards are set to 1 for correct responses and 0 for incorrect ones. We uniformly

adopt the token-mean loss aggregation and optimize all models 300 steps to convergence using AdamW (Loshchilov & Hutter, 2017) with a constant learning rate of $1 \times 10^{-6}$. All experiments are conducted on the VeRL (Sheng et al., 2025) framework using 8×A100 GPUs.

**Evaluation.** We comprehensively evaluate the effectiveness of these RL algorithms on seven competition-level mathematical reasoning benchmarks without contamination, including American Mathematics Competitions 2023 (AMC23), American Invitational Mathematics Examination 2024&2025 (AIME24, AIME25), Harvard—MIT Mathematics Tournament 2025 (HMMT25), Brown University Math Olympiad 2025 (BRUMO25), Carnegie Mellon Informatics and Mathematics Competition 2025 (CMIMC25), and Stanford Math Tournament 2025 (SMT25) (Balunović et al., 2025). We repeat the evaluation set for 8 times and report avg@8 metric for results stability.

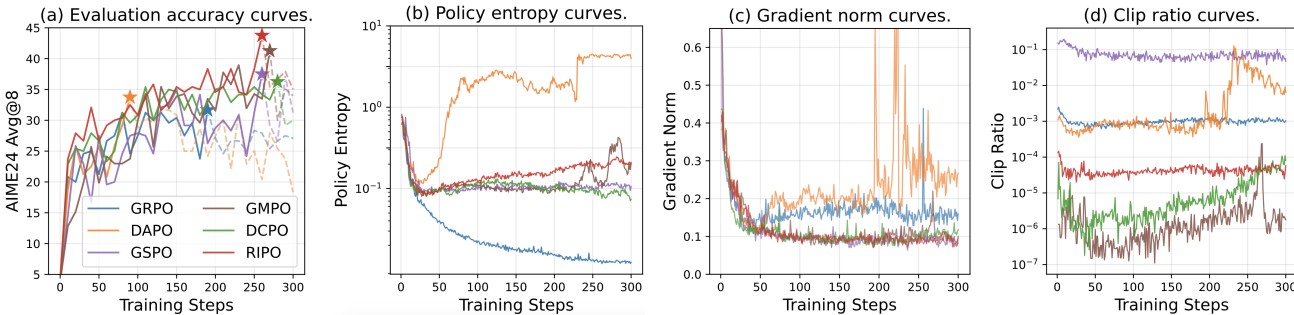

*Figure 1.* Training dynamics of Qwen3-8B-Base with various RL algorithms trained on DAPO-Math-17k.

## 5.2. Main Results

Table 1 summarizes the main experimental results across seven competition-level mathematical reasoning benchmarks and four base models with different scales and architectures.

Overall, RIPO consistently achieves the best average performance across all models, demonstrating strong effectiveness and generalization. Specifically, RIPO substantially outperforms GRPO up to 37.2%, 34.4%, 17.1%, and 35.1% on average for Qwen3-1.7B-Base, Llama3.2-3B-Instruct, Qwen3-4B-Base, and Qwen3-8B-Base, respectively, significantly improving the reasoning capability of different models. When compared to algorithms with improved clippings (*i.e.*, DAPO, GSPO, GMPO, and DCPO), RIPO leads consistently on the majority of benchmarks, especially on harder datasets like AIME24 and BRUMO25. The performance improvement is particularly notable on larger models, indicating that RIPO scales effectively with model size.

In summary, the experimental results demonstrate that RIPO delivers consistent and scalable improvements across diverse models and benchmarks, validating its theoretical foundation and practical effectiveness.

## 5.3. Training Dynamics and Analysis

We visualize the training dynamics of Qwen3-8B-Base with the six RL algorithms in Fig. 1.

As shown in Fig. 1(a), RIPO exhibits substantially faster performance improvement on AIME24 compared to other methods. Notably, RIPO surpasses the performance of GRPO trained for 200 steps within only 40 steps, demonstrating **five times the token-efficiency**. Moreover, the evaluation curve of RIPO increases smoothly with minimal oscillation and does not exhibit training collapse, indicating stable and efficient optimization.

Fig. 1(b) reports the policy entropy during training. GRPO suffers from a rapid entropy collapse to near zero, reflecting deterministic policy outputs and severe exploration collapse. While DAPO exhibits uncontrolled entropy growth,

indicating excessive exploration. In contrast, RIPO strikes a favorable balance between exploration and exploitation, its entropy decreases initially and then stabilizes within a moderate range, maintaining sustained exploration without destabilizing training.

Fig. 1(c) presents the gradient norm dynamics. Other RL algorithms show pronounced oscillations with frequent spikes, reflecting unstable updates and high-variance gradients during optimization. By contrast, RIPO maintains an almost fluctuation-free gradient norm throughout training, consistent with previous analysis that RIPO stabilizes optimization by balancing bias and variance.

Fig. 1(d) shows the proportion of clipped tokens. DCPO and GMPO rarely activate clipping, whereas GSPO and DAPO clip several orders of magnitude more tokens. In comparison, RIPO achieves a balanced clipping ratio between these extremes, indicating an appropriate and well-calibrated trust region that harmonize update flexibility and stability.

## 5.4. Ablation Study

We conduct an ablation study on the choice of $\delta$ for RIPO using Qwen3-8B-Base. In addition to the default symmetric setting, we also consider a decoupled variant, where different budgets are assigned to the lower and upper clipping boundaries, denoted as $\delta_{\text{low}}$ and $\delta_{\text{high}}$, respectively. Table 2 summarizes the results on AIME24. RIPO exhibits stable performance across a broad range of $\delta$ values from 0.02 to 0.08, demonstrating the robustness to hyper-parameter. In contrast, when $\delta_{\text{high}}$ is set substantially larger than $\delta_{\text{low}}$, performance degrades markedly, highlighting the importance of jointly constraining both directions of policy updates. Fig. 2 further illustrates the training dynamics. For symmetric settings, the reward improves steadily as entropy gradually decreases and stabilizes at a moderate level, indicating well-behaved optimization. In contrast, highly asymmetric settings exhibit sudden reward degradation with entropy explosion. This phenomenon arises because the asymmetric clipping makes action probabilities much easier to increase than to decrease, leading to excessive exploration and even-

*Table 2.* Ablation study of different $\delta$ for RIPO on AIME24.

| $\delta_{\text{low}}$ | 0.02 | 0.05 | 0.05 | 0.08 | 0.05 | 0.08 | 0.08 |
| $\delta_{\text{high}}$ | 0.02 | 0.04 | 0.05 | 0.08 | 0.02 | 0.02 | 0.04 |
| Avg@8 | 40.8 | 41.7 | **43.8** | 42.1 | 28.8 | 27.5 | 27.9 |

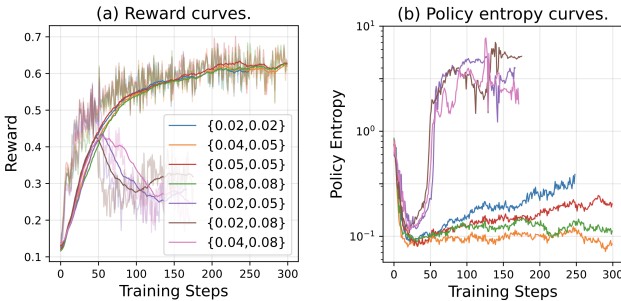

*Figure 2.* Training dynamics of RIPO with different $\{\delta_{\text{low}}, \delta_{\text{high}}\}$.

*Table 3.* Comparison of RL algorithms with different clipping motivations on Qwen3-8B-Base.

| Benchmark | GRPO | GPPO | Clip-Cov | **RIPO** |
|---|---|---|---|---|
| AIME24 | 31.7 | 31.7 (+0.0) | 36.3 (+4.6) | **43.8 (+12.1)** |
| AIME25 | 20.8 | 23.8 (+3.0) | 22.9 (+2.1) | **29.2 (+ 8.4)** |
| AMC23 | 66.6 | 73.4 (+6.8) | 66.6 (+0.0) | **79.7 (+13.1)** |
| HMMT25 | 12.9 | 14.2 (+1.3) | 11.7 (-1.2) | **16.7 (+ 3.8)** |
| BRUMO25 | 33.3 | 38.3 (+5.0) | 36.3 (+3.0) | **47.5 (+14.2)** |
| SMT25 | 21.9 | 25.0 (+3.1) | 30.2 (+8.3) | **34.2 (+14.2)** |

tually causes training collapse.

## 5.5. Comparison with Other Clippings

We further compare RIPO with RL algorithms that employ clipping mechanisms with different underlying motivations and strategies. Specifically, we investigate GPPO (Su et al., 2025) which preserves the gradients of PPO-Clipped tokens to mitigate information loss, and Clip-Cov (Cui et al., 2025b) which clips these high-covariance tokens to regulate entropy and encourage exploration. We conduct experiments on the Qwen3-8B-Base model, following the previous experimental settings and using the default hyperparameters for each method. Table 3 summarizes the quantitative results. While GPPO and Clip-Cov show improvement over GRPO, RIPO significantly outperforms them across all benchmarks, highlighting the superior stability and effectiveness.

## 5.6. Transfer to PPO Objective

To demonstrate the generality of RIPO-Clip (*i.e.*, RIC) beyond GRPO-style objectives, we further apply it to PPO objective. Compared with the group-relative RL algorithms studied above, PPO differs in that it employs a learned value

*Table 4.* Comparison of different clipping mechanisms on PPO objective (evaluated on GSM8k dataset by Avg@1).

| Models | PPO-Clip | DAPO-Clip | DCPO-Clip | **RIPO-Clip** |
|---|---|---|---|---|
| 0.5B | 58.0 (+0.0) | 58.7 (+0.7) | 58.4 (+0.4) | **61.1 (+3.1)** |
| 1.5B | 79.2 (+0.0) | 80.9 (+1.7) | 79.6 (+0.4) | **81.6 (+2.4)** |
| 7B | 91.7 (+0.0) | 90.8 (-0.9) | 91.4 (-0.3) | **93.5 (+1.8)** |
| 14B | 93.2 (+0.0) | 93.6 (+0.4) | 93.9 (+0.7) | **94.4 (+1.2)** |

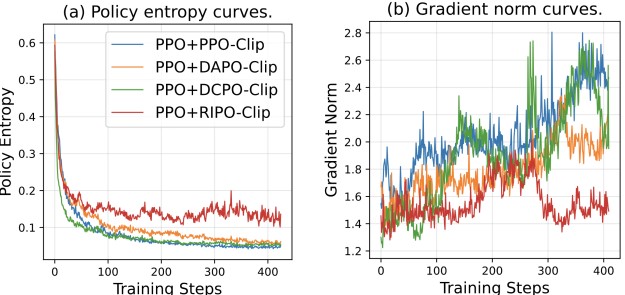

*Figure 3.* Training dynamics of Qwen2.5-1.5B-Instruct with different clipping mechanisms on GSM8k.

function and uses generalized advantage estimation (GAE) to compute advantages, as described in Sec. 2.

We conduct experiments on the GSM8K math dataset (Cobbe et al., 2021), consisting of 7K training problems and 1K held-out test problems. We evaluate Qwen2.5-Instruct models (Qwen et al., 2025) of different scales under the PPO objective, and compare RIC with other clipping mechanisms. The maximum response length is set as 8,192 tokens for thorough reasoning. In each RL iteration, the behavior policy $\pi_{\theta_{old}}$ totally generates 256 rollouts with a train batch size of 512, and the current policy $\pi_\theta$ is updated 2 times with a mini-batch size of 256. For the value model of PPO, we adopt the same model as policy does with no warmups. We optimize all models 15 epochs (435 steps) to convergence using AdamW with a learning rate of $1e-6$.

Table 4 reports the results measured by Avg@1. Despite the difference in objective formulation and advantage estimation, RIPO-Clip consistently outperforms other three clipping mechanisms, demonstrating that RIC is broadly applicable across policy optimization frameworks.

Fig. 3 visualizes the training dynamics of Qwen2.5-1.5B-Instruct with different clips. PPO-Clip encounters severe exploration collapse where entropy rapidly decays to near zero. While PPO with RIPO-Clip maintains in certain entropy level, promoting consistent exploration. Additionally, the gradient norm of PPO with other clips exhibits substantial fluctuations and sharp spikes. While RIPO-Clip demonstrates smoother gradients, signifying more stable optimization.

*Table 5.* Pass@k performance comparison of different methods based on Qwen3-8B-Base evaluated on AIME-25 and HMMT-25

| Dataset | Mehthod | Avg@8 | Pass@1 | Pass@8 | Pass@16 | Pass@32 | Pass@64 | Pass@128 |
|---------|---------|-------|--------|--------|---------|---------|---------|----------|
| AIME-25 | Base | 3.3 | 5.0 | 12.3 | 16.7 | 16.7 | 16.7 | 16.7 |
| | GRPO | 20.8 | 20.4 | 36.5 | 40.8 | 45.5 | 50.1 | 53.3 |
| | DAPO | 22.1 | 20.6 | 33.4 | 35.7 | 37.4 | 39.1 | 40.0 |
| | DCPO | 27.5 | 26.4 | 39.5 | 44.7 | 48.5 | 54.7 | 58.9 |
| | RIPO | 29.2 | 30.4 | 43.2 | 47.0 | 50.8 | 55.6 | 60.0 |
| HMMT-25 | Base | 1.3 | 0.6 | 3.5 | 5.2 | 6.3 | 6.7 | 6.7 |
| | GRPO | 12.9 | 11.7 | 19.3 | 21.6 | 24.1 | 26.6 | 30.0 |
| | DAPO | 7.9 | 7.7 | 16.7 | 20.3 | 23.4 | 25.7 | 26.7 |
| | DCPO | 15.4 | 15.5 | 26.8 | 30.6 | 34.3 | 37.5 | 40.0 |
| | RIPO | 16.7 | 16.7 | 28.3 | 32.6 | 37.3 | 41.4 | 45.3 |

*Table 6.* Comparison of different RL algorithms (evaluated by Avg@8) on various coding and search tasks

| Task Type | Model | Codeforces | CodeContest | TACO | APPS | Average |
|-----------|-------|------------|-------------|------|------|---------|
| Coding | Base | 20.2 | 25.8 | 10.4 | 25.8 | $20.6\,(-48.1\%)$ |
| | GRPO | 46.8 | 44.8 | 19.7 | 47.6 | $39.7\,(+\ 0.0\%)$ |
| | RIPO | 51.5 | 49.9 | 25.8 | 52.5 | $44.9\,(+13.2\%)$ |

| Task Type | Model | TriviaQA | PopQA | HotpotQA | WikiMultiHopQA | Average |
|-----------|-------|----------|-------|----------|----------------|---------|
| Search | Base | 19.5 | 7.2 | 9.3 | 17.7 | $13.4\,(-64.5\%)$ |
| | GRPO | 60.9 | 34.5 | 25.1 | 30.4 | $37.7\,(+\ 0.0\%)$ |
| | RIPO | 67.4 | 39.8 | 30.9 | 35.5 | $43.4\,(+15.1\%)$ |

## 5.7. RIPO Breaks through the Capacity Boundaries

We additionally conduct deep-dive Pass@k analysis (up to k=128) of Qwen3-8B-Base on AIME-25 and HMMT-25, the two most challenging benchmarks requiring complex reasoning, to study the capacity boundaries of LLMs trained with different RL algorithms.

As Table 5 shows, the base model's performance plateaus prematurely around $k = 16$, suggesting a limited intrinsic capacity. In contrast, RIPO consistently outperforms other RL algorithms and scales continuously, achieving a peak of 60.0% on AIME-25 and 45.3% on HMMT-25 at $k = 128$. These compelling results empirically demonstrate that RIPO effectively mitigates the severe exploration collapse issue in long-horizon reasoning tasks, successfully maintaining policy diversity and breaking through the intrinsic boundaries of reasoning capacity imposed by the base model.

## 5.8. Generalization to Coding and Search Tasks

We further extend to long-horizon coding and multi-hop search tasks. For coding, we trained on Eurus-Code dataset (Cui et al., 2025a) and evaluated on four challenging benchmarks, namely, Codeforces, CodeContest, TACO (Text-Assisted Coding with Objectives), and APPS (Automated Programming Progress Standard). For search, we

trained on Search-R1 dataset (Jin et al., 2025) and evaluated on TriviaQA, PopQA, HotpotQA, and WikiMultiHopQA. We conduct experiments on Qwen3-8B-Base following previous stated training recipes.

As Table 6 shows, RIPO consistently outperforms GRPO with significant gains across these benchmarks, validating the universal effectiveness of our method across different long-horizon reasoning tasks, illustrating the geometric mismatch we addressed is intrinsic and critical.

## 6. Conclusion

This paper theoretically identifies that the fundamental limitation of PPO-Clip stems from the mismatch between the Euclidean metric and the intrinsic geometry of the policy manifold. Grounded in this derivation, we introduce Riemannian Isometric Policy Optimization (RIPO), which guarantees isometric policy updates on the manifold to balance exploration and exploitation. This geometric isometry also induces statistical homoscedasticity, enabling a favorable bias-variance trade-off. Extensive experiments on multiple competition-level benchmarks show that RIPO consistently and substantially outperforms representative RL baselines. Our work establishes a principled pathway for designing more effective and stable reinforcement learning algorithms for large language models.

## Acknowledgments

This work is supported by the Natural Science Foundation of China (Grant No. 62376133) and sponsored by Beijing Nova Program (20240484682) and the Wuxi Research Institute of Applied Technologies, Tsinghua University (20242001120).

## Impact Statement

This paper introduces Riemannian Isometric Policy Optimization (RIPO), a theoretically grounded RL algorithm that resolves a fundamental flaw underlying clipping-based policy optimization. By enforcing geometry-isometric policy updates, RIPO establishes a principled framework for stable and scalable RL. More broadly, this work delivers valuable theoretical insights for the design of RL algorithms and significantly enhances the reasoning capabilities of large language models on addressing more challenging tasks.

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
