# OpenReview forum: "Beyond Euclidean Clipping: Overcoming Exploration Collapse in LLM RL via Riemannian Isometric Policy Optimization"
_ICML.cc/2026/Conference — ICML 2026 regular_

### Official Review · Reviewer_GjsB · 2026-02-26

**Soundness:** 3
**Presentation:** 3
**Significance:** 3
**Originality:** 3
**Overall Recommendation:** 4
**Confidence:** 3

**Summary:**

This paper proposes RIPO, a PPO-based RL training approach with Riemannian Geometry, which aims to mitigate the insufficient exploration capability of the PPO-clip in large reasoning models. The authors analyze that the flaw of PPO-clip arises from the inconsistency between the Euclidean metric and the intrinsic geometric structure on the policy Riemannian manifold, and accordingly introduce RIC as the clipping strategy. Empirically, extensive experimental results demonstrate the effectiveness and stable optimization of RIPO.

**Compliance With Llm Reviewing Policy:**

Affirmed.

**Final Justification:**

The rebuttal has addressed my concerns. I'll maintain my positive score.

**Key Questions For Authors:**

* For the first-order Taylor expansion of Eq.(6), should it be an approximation?
* Regarding Figure 1 (a), why do all the curves drop sharply after reaching their peak?

**Limitations:**

It is unclear whether the proposed methods can be successfully applied to other domains, such as RLHF or code generation.

**Strengths And Weaknesses:**

Strengths:
* This paper explains the insufficient exploration capability of PPO-clip by introducing Riemannian geometry.
* This paper proposes a simple and effective method RIPO, which enables stable policy optimization.
* This paper demonstrates the effectiveness of adopting RIC by analyzing it through the bias-variance trade-off in off-policy RL.

Weakness:
* The number of references is insufficient, and there is a lack of introduction and discussion of related work, especially on LLM RL beyond PPO-clip.
* This paper adopts Avg@8 as the evaluation metric, which is not the standard metric employed in prior work. To ensure comparability, it is recommended that Pass@1 and Avg@32 be added as supplementary evaluation results.
* It is unclear about the performance of RIPO under standard RL tasks.

---

> ### Author Rebuttal · Authors · 2026-03-31
>
> We thank the reviewer for the insightful feedback and recognizing the significance and effectiveness of RIPO.
> We address the concerns as follows:
>
> ## Q1: Insufficient References
> We would like to clarify that in the last two paragraphs of Sec 2.3, we indeed introduce several modern LLM RL variants beyond PPO-clip, including DAPO, DCPO, GSPO, GMPO, CISPO, and GPPO.
> And we discuss that "They remain largely heuristic with limited theoretical understanding. Importantly, they do not identify and address the fundamental flaw of PPO-Clip". In contrast, our work  identifies the fundamental geometric flaw of PPO-Clip as the root cause of exploration collapse and proposes theoretically grounded algorithm RIPO.
>
> Constrained by page limits, the Preliminary Section in the main text focuses on core theoretical motivation.
> To provide a broader context of related work, we will add to the Appendix:
> * A Related Work section covering a broader landscape of LLM RL and providing more comprehensive context.
> * Detailed mathematical formulations and comparisons of existing RL algorithms.
>
> ## Q2: Evaluation Metrics
> We agree that Pass@1 and Pass@k are critical for comparability.
> To ensure a rigorous evaluation, we conducted a deep-dive analysis on AIME-25 and HMMT-25 with Qwen3-8B-Base using **128 rollouts** per question. This provides a more stable estimation of Pass@1 (estimated as the mean of 128 samples) than 32 rollouts.
> | AIME-25 | Pass@1 | Pass@8 |  Pass@32 | Pass@64 | Pass@128 |Avg@8 | Avg@128 |
> | :---|  :---: |  :---: | :---: |  :---: |  :---: | :---: |  :---: |
> | Base    | 5.0 | 12.3 | 16.7 | 16.7 | 16.7 | 3.3 | 5.0 |
> | GRPO | 20.4 | 36.5 |45.5 | 50.1 | 53.3 | 20.8 | 20.4 |
> | **RIPO**    | 30.4 | 43.2 | 50.8 | 55.6 | 60.0 | 29.2 | 30.4 |
>
> | HMMT-25  | Pass@1 | Pass@8 |  Pass@32 | Pass@64 | Pass@128 |Avg@8 | Avg@128 |
> | :---|  :---: |  :---: | :---: |  :---: |  :---: | :---: |  :---: |
> | Base   | 1.0 | 3.5 | 6.7 | 6.7 | 6.7 |  1.3 | 1.0 |
> | GRPO | 11.7 | 19.3 | 24.1 | 26.6 | 30.0 | 12.9 | 11.7 |
> | **RIPO**    | 16.7 | 28.3 | 37.3 | 41.1 | 45.3 | 16.7 | 16.7 |
>
> As observed, RIPO consistently outperforms GRPO across all standard metrics, effectively pushing the reasoning capacity boundaries of the base model.
> Moreover,  the high consistency between Avg@8 and Pass@1 (Avg@128) confirms that our original evaluation using Avg@8 was a reliable indicator of model performance.
> We will include a full Pass@k table in Appendix.
>
> ## Q3: Performance under Standard RL Tasks
> We evaluate the performance of RIPO under two widely-used standard RL tasks, Atari and MuJoCo, following the original PPO configurations:
> | Atari | PPO | RIPO |&#8593; |
> | :--- | :---: | :---: | ---: |
> | Breakout | 406.6 | **426.3** | +4.8% |
> | Pong | 20.2 | **22.4** | +10.9% |
> | BeamRider | 2445.4 | **2609.7** | +6.7% |
>
> | MuJoCo| PPO | RIPO |&#8593; |
> | :--- | :---: | :---: | :---: |
> | Hopper | 2231.1 | **2507.5** | +12.4% |
> | Walker2d | 3050.1 | **3312.8** | +8.6% |
> | HalfCheetah | 1822.8 | **2146.3** | +17.7% |
>
> RIPO consistently outperforms PPO across all tested environments with significant gain. The superior performances demonstrate that RIPO is not limited to LLM reasoning but effectively extends to traditional RL tasks, further validating the **theoretical robustness** and **universal effectiveness** of RIPO.
>
> ## Q4: Generalization to Coding and Search Tasks
> We conducted experiments on two challenging tasks, Coding and Multi-hop Search, using Qwen3-8B-Base.
>
> For coding, we train on Eurus-Code dataset, and evaluate on four challenging benchmarks, Codeforces, CodeContest, TACO (Text-Assisted Coding with Objectives), and APPS (Automated Programming Progress Standard). For search, we train on Search-R1 dataset, and evaluate on TriviaQA, PopQA, HotpotQA, and WikiMultiHopQA. We strictly followed the training recipes in paper and ensured that training/test sets had no overlap or contamination.
> The table below lists the results (Avg@8).
>
> | Coding | Codeforces | CodeContest | TACO | APPS | Average |
> | :--- | :---: | :---: | :---: | :---: | :---: |
> | Base | 20.2 | 25.8 | 10.4 | 25.8 | 20.6 (-48.1%) |
> | GRPO | 46.8 | 44.8 | 19.7 | 47.6 | 39.7 (+0.0%) |
> | **RIPO** | 51.5 | 49.9 | 25.8 | 52.5 | 44.9 (+13.2%) |
>
> | Search | TriviaQA | PopQA | HotpotQA | WikiMultiHopQA | Average |
> | :--- | :---: | :---: | :---: | :---: | :---: |
> | Base | 19.5 | 7.2 | 9.3 | 17.7 | 13.4 (-64.5%) |
> | GRPO | 60.9 | 34.5 | 25.1 | 30.4 | 37.7 (+0.0%) |
> | **RIPO** | 67.4 | 39.8 | 30.9 | 35.5 | 43.4 (+15.1%) |
>
> RIPO consistently outperforms GRPO with significant relative gains across these benchmarks.
> The results validate the **universal effectiveness** of our method across different long-horizon reasoning tasks, illustrating the geometric mismatch we address is **intrinsic and critical**.
>
> ***
>
> We hope our response and the additional evidence adequately address your concerns. We kindly invite you to reconsider the overall evaluation of our work.

---

> > ### Author Rebuttal · Reviewer_GjsB · 2026-04-03
> >
> > I sincerely thank the authors for their detailed and thoughtful response. My previous concerns have been adequately addressed, and I have no further comments. I will therefore maintain my positive evaluation of this manuscript.

---

> > > ### Author Response · Authors · 2026-04-08
> > >
> > > Dear Reviewer,
> > >
> > > We sincerely thank you for acknowledging that all concerns have been "fully resolved" and for your insightful suggestions which have made the paper stronger.
> > >
> > > We believe RIPO represents a high-impact advancement for LLM RL reasoning, combining profound theoretical depth with SOTA performance, and we look forward to its contribution to the field.
> > >
> > > Best regards,
> > >
> > > The Authors

---

### Official Review · Reviewer_1SVf · 2026-03-11

**Soundness:** 2
**Presentation:** 3
**Significance:** 2
**Originality:** 2
**Overall Recommendation:** 3
**Confidence:** 4

**Summary:**

This paper studies a failure mode of PPO-style reinforcement learning for large language models: exploration collapse caused by geometry mismatch. The core claim is that PPO-Clip measures policy updates with an Euclidean clipping rule even though discrete policies live on a probability simplex with non-Euclidean intrinsic geometry. The paper argues that this mismatch makes updates too conservative on low-probability actions and too aggressive on high-probability actions, which hurts exploration in reasoning-oriented RL. To address this, the authors introduce Riemannian Isometric Clip, which adapts the clipping boundary according to the local geometry of the simplex, and then build the full Riemannian Isometric Policy Optimization algorithm on top of it.

**Compliance With Llm Reviewing Policy:**

Affirmed.

**Final Justification:**

This rebuttal alleviated some of my concerns.

**Key Questions For Authors:**

1. Could the authors provide evidence that the same geometry-mismatch mechanism is a primary bottleneck outside mathematical reasoning RL, for example in preference optimization or agent-style post-training? A convincing answer would increase my confidence in the broader significance of the method.
2. How sensitive is RIPO to reward noise, advantage normalization, and value-estimation error compared with PPO-Clip? If the gains are robust under these common sources of variance, my soundness assessment would strengthen.
3. What is the practical implementation overhead of RIC/RIPO in large-scale distributed training, including memory, latency, and hyperparameter tuning cost?
4. Could the authors disentangle how much of the gain comes from better low-probability exploration versus smoother optimization dynamics more generally? A careful analysis here would clarify the mechanism and improve the paper's explanatory strength.

**Limitations:**

The paper motivates the domain well and empirically demonstrates benefits on reasoning RL, but it should discuss more explicitly that the current evidence is limited to a narrow task family and that the theoretical analysis abstracts away several training-system effects present in modern LLM RL pipelines.

**Strengths And Weaknesses:**

The paper presents a coherent technical story. The mismatch between Euclidean clipping and simplex geometry is well motivated, and the proposed RIC rule follows naturally from the goal of preserving a geometry-aware trust region. The empirical section is also relevant to the main claim: the authors test multiple base models and several competition-level reasoning benchmarks, include ablations on clipping parameters, and compare different clipping mechanisms.

However, the evidence is still somewhat narrow. The empirical domain is almost entirely mathematical reasoning RL, so it remains unclear whether the same geometric effect dominates in broader post-training settings involving preference learning, tool use, or agentic feedback. The paper technically solid but not fully exhaustive.

The paper is generally clear and well structured. The main intuition appears early, the illustrative trust-region example is useful, and the transition from diagnosis to algorithm design is easy to follow. The link between RIC and the full RIPO procedure is also clear. A remaining weakness is that some readers may want a more explicit operational comparison to standard PPO variants used in current LLM RL pipelines, especially around implementation overhead, interaction with KL regularization, and sensitivity to reward noise.

---

> ### Author Rebuttal · Authors · 2026-03-31
>
> We thank the reviewer for the insightful feedback and recognizing the "coherent technical story" and "principled motivation" of RIPO.
> We address the concerns below:
> ## Q1: Agent-style Tasks
> We train RIPO on two agent-style tasks, **Coding** (Eurus-Code) and **Multi-hop Search** (Search-R1), requiring agentic interaction with coding environment or search engine.
> We use Qwen3-8B-Base and follow the training recipes in paper. We evaluate on eight challenging benchmarks with no leakage or contamination. Table below lists the results (Avg@8).
> |Coding|Codeforces|CodeContest|TACO|APPS|Avg.|
> | :--- | :---: | :---: | :---: | :---: | :---: |
> |Base|20.2|25.8|10.4|25.8|20.6|
> |GRPO|46.8|44.8|19.7|47.6|39.7|
> |**RIPO** |51.5|49.9|25.8|52.5|44.9 (+13.2%)|
>
> |Search|TriviaQA|PopQA|HotpotQA|WikiMultiHopQA|Avg.|
> | :--- | :---: | :---: | :---: | :---: | :---: |
> |Base|19.5|7.2|9.3|17.7|13.4|
> |GRPO|60.9|34.5|25.1|30.4|37.7|
> |**RIPO** |67.4|39.8|30.9|35.5|43.4 (+15.1%)|
>
> RIPO’s gains validate its **universal effectiveness**, proving that the identified geometric mismatch is fundamental.
> ## Q2: Robustness to Noise & Errors
> Our extended experiments confirm RIPO maintains stability where PPO/GRPO fails under noise and errors.
> ### 1.Preliminary Experiments
> We first conduct preliminary experiments on GSM8K with Qwen2.5-0.5B-Instruct (PPO as baseline).
> * **Value-Estimation Error** is inherently exist in PPO baseline. RIPO's 61 surpasses PPO's 58, indicating robustness to estimation error.
> * **Advantage Normalization**: PPO drops to 57.7, while RIPO improves to 61.7, indicating compatibility.
> * **Reward Perturbations**. We test three types of perturbations, injecting Gaussian noise $\mathcal{N}(0, \sigma^2)$ with $\sigma=0.1$ and $\sigma=0.3$, and randomly flipping 10% 0/1 rewards.
>
> | |Base|+ Adv Norm|+ Noise  $\sigma=0.1$|+ Noise  $\sigma=0.3$|+10% Flip|
> | :--- | :---: | :---: |  :---: |  :---: |  :---: |
> |PPO|58.0|57.7|55.4|52.7|51.0|
> |**RIPO**|61.1|61.7|60.6|60.3|60.2|
>
> While PPO degrades significantly (-9% and -12%) under reward perturbations, RIPO remains remarkably resilient (only -1%).
> ### 2.Large-Scale Validation
> We further extend to DAPO-Math-17K with Qwen3-8B-Base using GRPO as baseline (evaluated on AIME-24). While GRPO deteriorates significantly, RIPO remains stable, providing compelling evidence of its robustness.
> | |Base|+ Noise  $\sigma=0.1$|+ Noise  $\sigma=0.3$|+10% Flip|
> | :--- | :---: | :---: |  :---: |  :---: |
> |GRPO|31.7|27.5|23.1|21.4|
> |RIPO|43.8|43.3|42.5|41.8|
> ### 3.Theoretical Insight
> As analyzed in Sec 3.1&3.2, PPO-Clip often suffers from pathological over-updates. Thus an erroneous reward can make the policy take an excessively large step in the wrong direction, leading to irreversible divergence. In contrast, RIPO-Clip ensures every update within a geometrically consistent trust region, thus the magnitude of erroneous update is strictly constrained, allowing the **global optimization** trajectory to **remain correct**.
> ### 4.Traditional RL Tasks
> Traditional RL tasks (Atari and MuJoCo) are characterized by high-variance rewards.
> Results (please refer to Q3 of Rebuttal to Reviewer GjsB) show that RIPO consistently outperforms PPO, further validating the universal robustness and effectiveness of RIPO.
> ## Q3: Implementation Overhead
> RIPO's overhead is **negligible** compared to PPO/GRPO:
> * **Zero Extra Memory**: RIPO utilizes buffer-existing old policy probabilities and $\delta$. No extra activation or params checkpoints.
> * **No Measurable Latency**: RIPO-Clip is $\mathcal{O}(1)$, whose cost is invisible relative to tremendous LLM forward/backward, KV-cache I/O, and All-Reduce.
> * **Robust Tuning**: RIPO is robust to $\delta$. Fixed $\delta=0.05$ brings superior results across all tasks, eliminating per-task tuning. RIPO is also stable across $\delta \in [0.02, 0.08]$ as shown in Table 2.
> ## Q4: Better Exploration v.s. Smoother Optimization
> In RIPO, better exploration and smoother optimization are intrinsically coupled:
> * **Low-Prob. Exploration (Capacity)**: In complex tasks, correct paths often have low initial likelihoods. RIPO prevents these rare and vital gradients from being suppressed by PPO-Clip. This "geometric protection" empowers policy to explore rare trajectories and reach higher performance ceilings.
> * **Smoother Optimization (Stability)**: RIPO’s isometric properties ensure stable parameter updates, avoiding the pathological behavior   of PPO-Clip. Smoothness prevents policy collapse during high-difficulty exploration by mitigating variance and over-updates, ensuring the policy to achieve the performance ceiling.
> * **Synergistic Effect.** RIPO’s gains stem from this synergy, smoother optimization safely unleashes the model’s reasoning potential, while better exploration prevents stable optimization from trapping in mediocre local optima.
> ***
> We hope our response and the additional evidence adequately address your concerns. We kindly invite you to reconsider the overall evaluation of our work.

---

> > ### Author Rebuttal · Reviewer_1SVf · 2026-04-04
> >
> > Thank you for the thoughtful rebuttal and for providing additional experiments on agent-style tasks, robustness to reward perturbations, and implementation overhead. These additions are helpful and make the empirical picture broader than in the original submission.
> >
> > However, my concerns are only partially resolved. While the added coding/search results and robustness tests strengthen the paper, I still find the evidence somewhat limited relative to the breadth of the paper’s broader claims.
> >
> > Overall, the rebuttal improves the paper and addresses several points. I will increase my score.

---

> > > ### Author Response · Authors · 2026-04-05
> > >
> > > We sincerely thank the reviewer for the acknowledgment and for increasing the score. Regarding the remaining concern about the "limited evidence relative to broader claims," we would like to provide further context on why **we believe the current evidence is exceptionally rigorous and comprehensive within our work's scope of LLM Reasoning**.
> > >
> > > ### **1.RIPO Offers Broader Evaluation Compared to Existing SOTA Works**
> > >
> > > While many recent top-tier RL works for LLM Reasoning (e.g., GRPO [1] in Nature, DAPO [2], ORZ [3], CPPO [4] in NeurIPS '25, GMPO [5]  ICLR '26 and VinePPO [6] in ICML'25) **only** focus on **Math** as their empirical domain, RIPO has been validated across **three** distinct and high-impact domains: **Math, Coding and Search**.
> > >
> > > Moreover, we evaluate on various benchmarks, including seven benchmarks for Math, four benchmarks for coding and four benchmarks for search, **exceeding the current community standard for empirical proof in algorithmic contributions**.
> > >
> > > We have persuasively validated the **superior performance** of RIPO, which is the **SOTA** RL algorithm for improving the LLM reasoning capacity,  averagely surpassing GRPO by 30% in Math and 15% in Coding/Search.
> > > We think the experiments persuasively **validate the broad impact** of RIPO and are  **totally sufficient for a qualified top-tier conference paper**.
> > >
> > > ### **2.Reasoning as the Core Frontier for Exploration**
> > >
> > > RIPO is proposed to improve LLM reasoning capacity, which is the **core claim** of our paper. Math and Coding are the two best testbeds for evaluating the reasoning capacity of RL-trained LLMs due to their **long-horizon reasoning** settings. In these settings, LLMs **require extensive exploration** to find successful reasoning trajectories, thereby breaking through their reasoning capacity ceilings.
> > >
> > > Exploration is **vital** for modern reasoning-oriented LLM RL. However, as pointed out in our paper, **exploration collapse** is a critical issue in existing RL algorithms that limits reasoning capacity, and **no previous work** has identified the fundamental reason for this. Our work **theoretically identifies that the fundamental reason** stems from a geometric flaw and proposes RIPO to rectify this flaw. Thus, the contribution of our work to the realm of LLM reasoning is **highly significant**.
> > >
> > > In addition, the preference alignment mentioned by the reviewer is actually **out of our scope**. For preference alignment, the main target is to model human preferences, where exploitation is the primary focus instead of exploration. This vein of work mainly proceeds in a DPO-like manner, rather than a GRPO-like one.
> > >
> > > ### **3.RIPO has Theoretical Depth Combined with SOTA Performance**
> > >
> > > RIPO is not a heuristic trick but a fundamental theoretical advancement in reasoning LLM RL. While most existing works (PPO, GRPO, DAPO, etc.) rely on heuristic methods. RIPO is a **rare and principled** work:
> > > * **Theoretical Insight**: We theoretically identify that exploration collapse stems from the geometric mismatch between the Euclidean metric and the Riemannian manifold, as stated in our paper.
> > > * **Theoretical Grounding**: RIPO is a mathematically grounded algorithm to rectify this geometric flaw.
> > > * **Empirical Dominance**: RIPO achieves SOTA across all three reasoning domains without additional parameters or memory overhead, significantly surpassing existing reasoning LLM RL algorithms.
> > >
> > > To conclude, **we believe our work's quality and potential impact highly align with the highest standards of top-tier conferences.**
> > >
> > > ---
> > >
> > > We have made every effort to exceed the typical empirical requirements for an algorithmic contribution. Given the principled foundation, SOTA performance across three major domains, and proven robustness, we hope the reviewer agrees that RIPO represents a complete, high-quality, and highly significant contribution to the Reasoning LLM RL community.
> > >
> > > [1] DeepSeek-R1 incentivizes reasoning in LLMs through reinforcement learning (Nature 2025)
> > >
> > > [2] DAPO: An Open-Source LLM Reinforcement Learning System at Scale (NeurIPS 2025)
> > >
> > > [3] Open-Reasoner-Zero: An Open Source Approach to Scaling Up Reinforcement Learning on the Base Model (NeurIPS 2025)
> > >
> > > [4] CPPO: Accelerating the Training of Group Relative Policy Optimization-Based Reasoning Models (NeurIPS 2025)
> > >
> > > [5] Geometric-Mean Policy Optimization (ICLR 2026)
> > >
> > > [6] VinePPO: Refining Credit Assignment in RL Training of LLMs (ICML 2025)

---

### Official Review · Reviewer_e1oR · 2026-03-13

**Soundness:** 3
**Presentation:** 2
**Significance:** 2
**Originality:** 3
**Overall Recommendation:** 4
**Confidence:** 3

**Summary:**

The paper argues that PPO-Clip implicitly assumes a Euclidean metric on the importance ratio, which is inconsistent with the intrinsic Riemannian geometry induced by the KL divergence on the policy manifold. Based on this observation, it proposes Riemannian Isometric Policy Optimization (RIPO), which introduces a distribution-dependent clipping boundary that adapts to the local probability simplex. This mechanism allows larger updates for low-probability actions and smaller updates for high-probability actions, enabling balanced exploration and exploitation.

**Compliance With Llm Reviewing Policy:**

Affirmed.

**Final Justification:**

The authors' reply has addressed my main concerns.

**Key Questions For Authors:**

See weakness

**Limitations:**

No, more discussion on the potential problem of RIPO should be presented.

**Strengths And Weaknesses:**

Strength:
1. The paper identifies a geometric interpretation of PPO-Clip and attributes exploration collapse to a mismatch between Euclidean ratio clipping and KL-induced Riemannian geometry. This perspective is interesting and potentially useful for understanding RL optimization.

2. RIPO introduces a dynamic clipping boundary depending on policy probability, which is straightforward to implement within existing PPO/GRPO frameworks.

3. The derivation connecting KL geometry, Fisher information, and clipping behavior provides a principled motivation for the proposed method.

Weakness:

1.  More RL tasks, like Atari games and MuJoCo robot locomotion tasks, should be included for efficiency validation for RIPO.

2. Experiments focus mainly on math reasoning benchmarks; it remains unclear whether the improvements generalize to broader RLHF or alignment tasks.

4. More ablation studies (e.g., scaling with model size or training length) would strengthen the empirical claims.

---

> ### Author Rebuttal · Authors · 2026-03-30
>
> We are encouraged that the reviewer recognizes RIPO as a "principled motivation" with an "interesting geometric interpretation" that is "potentially useful for understanding RL optimization".
> These insights align perfectly with our goal: addressing the fundamental geometric flaw of PPO-clip that leads to exploration collapse.
>
> Below we address the concerns:
>
> ## Q1: Validation on Traditional RL Tasks (Atari & MuJoCo)
>
> While our primary focus is enhancing the reasoning capabilities of modern large language models (LLMs), the geometric mismatch we identify is a fundamental issue inherent in the policy optimization process itself. To demonstrate RIPO’s universal effectiveness, we evaluated its performance on traditional RL benchmarks (Atari and MuJoCo) following the original PPO configurations:
> | Atari | PPO | RIPO |&#8593; |
> | :--- | :---: | :---: | ---: |
> | Breakout | 406.6 | **426.3** | +4.8% |
> | Pong | 20.2 | **22.4** | +10.9% |
> | BeamRider | 2445.4 | **2609.7** | +6.7% |
>
> | MuJoCo| PPO | RIPO |&#8593; |
> | :--- | :---: | :---: | :---: |
> | Hopper | 2231.1 | **2507.5** | +12.4% |
> | Walker2d | 3050.1 | **3312.8** | +8.6% |
> | HalfCheetah | 1822.8 | **2146.3** | +17.7% |
>
> As observed, RIPO **consistently outperforms** PPO across all tested environments with **significant gain**. This cross-domain success demonstrates that RIPO is not limited to LLM reasoning but effectively extends to fundamental RL control and perception tasks, further validating the **theoretical robustness** and **universal effectiveness** of proposed RIPO.
>
> ## Q2: Generalization to Coding and Search Tasks
> To further validate the universal effectiveness of our method, we conducted additional experiments on two distinct challenging LLM RL tasks, Coding and Multi-hop Search, using Qwen3-8B-Base.
>
> For coding, where agent must generate precise code, we trained on Eurus-Code dataset and evaluated on four challenging benchmarks, Codeforces, CodeContest, TACO (Text-Assisted Coding with Objectives), and APPS (Automated Programming Progress Standard).
> For search, where agent learns to autonomously invoke search engines to solve multi-step reasoning, we trained on Search-R1 dataset and evaluated on TriviaQA, PopQA, HotpotQA, and WikiMultiHopQA.
> We strictly followed the training recipes in paper and ensured that training/test sets had no overlap or contamination. Below list the results (Avg@8).
> | Coding | Codeforces | CodeContest | TACO | APPS | Average |
> | :--- | :---: | :---: | :---: | :---: | :---: |
> | Base | 20.2 | 25.8 | 10.4 | 25.8 | 20.6 (-48.1%) |
> | GRPO | 46.8 | 44.8 | 19.7 | 47.6 | 39.7 (+0.0%) |
> | **RIPO** | **51.5** | **49.9** | **25.8** | **52.5** | **44.9 (+13.2%)** |
>
> | Search | TriviaQA | PopQA | HotpotQA | WikiMultiHopQA | Average |
> | :--- | :---: | :---: | :---: | :---: | :---: |
> | Base | 19.5 | 7.2 | 9.3 | 17.7 | 13.4 (-64.5%) |
> | GRPO | 60.9 | 34.5 | 25.1 | 30.4 | 37.7 (+0.0%) |
> | **RIPO** | **67.4** | **39.8** | **30.9** | **35.5** | **43.4 (+15.1%)** |
>
> RIPO consistently outperforms GRPO with significant relative gains across all eight benchmarks.
> The results validate the **universal effectiveness** of our method across different long-horizon reasoning tasks, illustrating the geometric mismatch we address is intrinsic and critical.
>
> ## Q3: Ablation Studies
> We would like to clarify that extensive ablation studies regarding model scale and training length are already integrated into our manuscript:
> * **Scaling with Model Size**: We have evaluated RIPO across a wide range of model parameters to ensure its effectiveness is not scale-dependent. This includes 1.7B, 3B, 4B, and 8B models in Table 1, and 0.5B, 1.5B, 7B, and 14B models in Table 4 (where $n$B denotes a model with $n$ billion parameters). The consistent gains across these **eight different scales** (from 0.5B to 14B) persuasively demonstrate the robustness and scalability of our method.
> * **Training Length and Convergence**: In our experiments, all evaluated methods were trained for the same number of steps and reached full convergence. Consequently, further extending the training length leads to marginal performance changes. This is further supported by Figure 2 (Left), where the training reward curves for RIPO have clearly plateaued, indicating that the policy has stabilized and additional training steps would yield diminishing returns.
> * **Sample Efficiency**: To further illustrate the performance under different training lengths, Figure 1 of the paper provides the test accuracy on AIME-24 at different training length (steps). It shows that RIPO not only reaches a higher performance ceiling but also exhibits superior sample efficiency compared to GRPO. Specifically, RIPO surpasses the performance of GRPO trained for 200 steps within only 40 steps, demonstrating  **five times the token-efficiency**.
>
>
> ***
>
>
> We hope our response and the additional evidence adequately address your concerns. We kindly invite you to reconsider the overall evaluation of our work.

---

> > ### Author Rebuttal · Reviewer_e1oR · 2026-04-06
> >
> > Thanks for your reply. I will raise my score.

---

> > > ### Author Response · Authors · 2026-04-08
> > >
> > > Dear Reviewer,
> > >
> > > We sincerely thank you for acknowledging that all concerns have been "fully resolved" and for your insightful suggestions which have made the paper stronger.
> > >
> > > We believe RIPO represents a high-impact advancement for LLM RL reasoning, combining profound theoretical depth with SOTA performance, and we look forward to its contribution to the field.
> > >
> > > Best regards,
> > >
> > > The Authors

---

### Official Review · Reviewer_NVz2 · 2026-03-13

**Soundness:** 2
**Presentation:** 3
**Significance:** 3
**Originality:** 3
**Overall Recommendation:** 4
**Confidence:** 2

**Summary:**

This paper proposes RIPO, which mitigates exploration collapse for Reasoning LLMs trained by GRPO-like algorithms. The key observation is that the trust region constraint in TRPO can be theoretically linked to the behavior-policy-weighted PPO-Clip constraint. PPO-Clip uses a fixed clipping rule on the importance ratio, which the paper claims to be geometrically inconsistent with the KL-constraint and prevents larger updates on low-probability actions. Backed up by the theory, RIPO instead employs a distribution-aware threshold $\sqrt{\delta / \pi_{\theta_{old}}(a|s)}$, which prevents conservative updates on low-probability actions while limiting aggressive updates on high-probability actions. In addition to this exploration-exploitation trade-off, RIPO better deals with the bias-variance trade-off of the surrogate objective. Experimental results show improved reasoning ability and sample efficiency compared with other GRPO-like algorithms across several math reasoning tasks.

**Compliance With Llm Reviewing Policy:**

Affirmed.

**Final Justification:**

As commented in the Strength section, this work is well-motivated and well-written. And the exploration collapse problem is a fundamental research question in online RL, especially for GRPO-trained large reasoning LLMs. The additional clarification and experiments in the authors' rebuttal have addressed my concerns regarding the soundness. So I am generally positive about this work. However, due to my low confidence, I choose to maintain my current rating.

**Key Questions For Authors:**

See weaknesses.

**Limitations:**

Sea weaknesses.

**Strengths And Weaknesses:**

***Strengths:***

1. The paper is well-organized and easy to follow. I did not encounter difficulty in understanding the paper’s content.
2. Math notations are well-defined and consistent. Derivations in the main body are correct and compact without redundant information.
3. The paper’s motivation is important and intuitive. Exploration is a fundamental issue in RL, not limited to reasoning LLM.
4. The resulting RIPO algorithm is simple and effective, given the paper’s evaluation. Other metrics, like policy entropy and gradient norm, also reflect RIPO-policy’s stability and higher expressiveness.
---
***Weaknesses:***

Overall, it was an enjoyable read for me. I do not have major concerns about the paper’s content, but I am less familiar with the recent literature on the XXPO series for reasoning models. Despite this, I am curious about the following:

1. The theory relies on up to second-order Taylor expansion of $\pi\_{\theta}(a|s)$ around $\theta\_{old}$. Have the authors evaluated whether the RIPO update remains sufficiently local for this approximation to be valid by not deviating too much from old parameters? How about the actual KL and the approximated KL during training?
2. People like to discuss Reasoning LLM capacity and accuracy [1]. How would RIPO behave differently from baselines in terms of pass@k score?
3. Evaluations are only done on math reasoning tasks. How about other long-horizon reasoning tasks? Nowadays, I think related work is usually evaluated on multiple domains.

***Minor:***

1. I think some statistical terminologies are too strong or general to use in this paper’s context. The authors may consider relaxing them to prevent the risk of overclaiming.
2. In Left Line 071, it should be “balancing exploration and exploitation…”
---
***References:***

[1] Does Reinforcement Learning Really Incentivize Reasoning Capacity in LLMs Beyond the Base Model? (2025)

---

> ### Author Rebuttal · Authors · 2026-03-30
>
> We thank the reviewer for the encouraging feedback and recognizing RIPO as an "enjoyable read" with "important motivation".
> We address the reviewer's concerns below:
>
> ## Q1: KL Approximation and Locality
> We will clarify the KL approximation and sufficient locality from both theoretical and empirical perspectives:
> * **Theoretical Orthodoxy**: RIPO is a principled restoration of the TRPO framework, recovering its second-order essence within a proximal setting. While PPO provides a crude first-order simplification that ignores local curvature, RIPO’s local approximation offers a more rigorous second-order trust-region constraint. Besides, TRPO also derives the surrogate objective via the same local approximation of $\theta$ around $\theta_{old}$, thus RIPO's local approximation is theoretically orthodox.
> * **Empirical Validation**: Our training logs confirm that RIPO's actual KL (\~0.00015 with flat curve during training) is consistently lower and more stable than GRPO's (\~0.00025 with spikes and large fluctuations) under the same budget. This proves that distribution-aware  clipping anchors the policy within the trust region more effectively than fixed clipping of PPO. Since the actual KL remains within a minimal range, the locality assumption for the second-order Taylor expansion is strictly satisfied.  We will add the KL curves in the revised Appendix.
>
>
> ## Q2: Pass@K and Capacity Boundaries
> We additionally conduct deep-dive Pass@k analysis (up to k=128) of Qwen3-8B-Base on AIME-25 and HMMT-25, the two most challenging benchmarks requiring complex reasoning.
> | AIME-25 | Avg@8 | Pass@1 | Pass@8 | Pass@16 | Pass@32 | Pass@64 | Pass@128 |
> | :--- | :---: |  :---: |  :---: | :---: | :---: |  :---: |  :---: |
> | Base   | 3.3 | 5.0 | 12.3 | 16.7 | 16.7 | 16.7 | 16.7 |
> | GRPO | 20.8 | 20.4 | 36.5 | 40.8 | 45.5 | 50.1 | 53.3 |
> | **RIPO**   | **29.2** | **30.4** | **43.2** | **47.0** | **50.8** | **55.6** | **60.0** |
>
> | HMMT-25 | Avg@8 | Pass@1 | Pass@8 | Pass@16 | Pass@32 | Pass@64 | Pass@128 |
> | :--- | :---: |  :---: |  :---: | :---: | :---: |  :---: |  :---: |
> | Base   | 1.3 | 1.0 | 3.5 | 5.2 | 6.7 | 6.7 | 6.7 |
> | GRPO | 12.9 | 11.7 | 19.3 | 21.6 | 24.1 | 26.6 | 30.0 |
> | **RIPO**   | **16.7** | **16.7** | **28.3** | **32.6** | **37.3** | **41.1** | **45.3** |
>
> The base model’s performance plateaus prematurely (around k=16), suggesting a limited intrinsic capacity.
> In contrast, RIPO consistently scales, achieving 60.0% on AIME-25 at k=128.
> This proves RIPO effectively fixes exploration collapse and **breaks through the boundaries of reasoning capacity** beyond the base model.
> We will include a full Pass@k table in the revised Appendix.
>
>
> ## Q3: Experiments on Coding and Search Tasks
> To address the concern regarding task diversity, we evaluated RIPO on **long-horizon coding and multi-hop search tasks**.
> For coding, we trained Qwen3-8B-Base on Eurus-Code dataset[1], and evaluated on four challenging benchmarks, Codeforces, CodeContest, TACO (Text-Assisted Coding with Objectives), and APPS (Automated Programming Progress Standard).
> For search, we trained Qwen3-8B-Base on Search-R1 dataset[2], and evaluated on TriviaQA, PopQA, HotpotQA, and WikiMultiHopQA. We strictly followed the training recipes in our paper and ensured that training/test sets had no overlap or contamination.
> The table below lists the results (Avg@8):
> | Coding | Codeforces | CodeContest | TACO | APPS | Average |
> | :--- | :---: | :---: | :---: | :---: | :---: |
> | Base | 20.2 | 25.8 | 10.4 | 25.8 | 20.6 (-48.1%) |
> | GRPO | 46.8 | 44.8 | 19.7 | 47.6 | 39.7 (+0.0%) |
> | **RIPO** | **51.5** | **49.9** | **25.8** | **52.5** | **44.9 (+13.2%)** |
>
> | Search | TriviaQA | PopQA | HotpotQA | WikiMultiHopQA | Average |
> | :--- | :---: | :---: | :---: | :---: | :---: |
> | Base | 19.5 | 7.2 | 9.3 | 17.7 | 13.4 (-64.5%) |
> | GRPO | 60.9 | 34.5 | 25.1 | 30.4 | 37.7 (+0.0%) |
> | **RIPO** | **67.4** | **39.8** | **30.9** | **35.5** | **43.4 (+15.1%)** |
>
> RIPO consistently outperforms GRPO with **significant gains** across these benchmarks.
> The results validate the **universal effectiveness** of our method across different long-horizon reasoning tasks, illustrating the geometric mismatch we address is **intrinsic and critical**.
>
>
> ## Q4: Minor points
> We will relax statistical terms to avoid overclaiming, and correct the typo in L071.
>
>
> ***
>
>
> We hope our response and the additional evidence adequately address your concerns. We kindly invite you to reconsider the overall evaluation of our work.
>
> [1] Process Reinforcement through Implicit Rewards
>
> [2] Search-R1: Training LLMs to Reason and Leverage Search Engines with Reinforcement Learning

---

> > ### Author Rebuttal · Reviewer_NVz2 · 2026-04-03
> >
> > I confirm that I have read the rebuttal and other reviews. I thank the authors for their organized response to my review. The additional clarification and experiments have addressed my concerns. Due to my low confidence, I choose to maintain my current rating. But I am positive about this manuscript.
> >
> > I noticed that the authors did some experiments with MuJoCo and Atari. I am curious: (1) how many seeds were used for each experiment; (2) how many environment steps were trained; (3) which MuJoCo version was chosen (e.g., Hopper-v4)?

---

> > > ### Author Response · Authors · 2026-04-08
> > >
> > > Dear reviewer, we sincerely thank you for acknowledging that all concerns have been "fully resolved" and for your positive assessment.
> > >
> > > To further bolster your confidence in the **significance and impact** of RIPO,  we would like to provide additional context highlighting our **theoretical depth and superior performance** compared to existing top-tier works in LLM RL for reasoning.
> > >
> > > ### **1.First-Principled Advancement vs. Heuristic Modifications**
> > >
> > > While existing SOTA methods have made significant empirical strides, they  remain **largely heuristic** in nature, focusing on hyperparameter tuning or formulation tweaks without addressing the underlying theoretical flaw of PPO/GRPO. In contrast, **RIPO is a first-principled advancement with theoretical depth that rectifies a decade-old and unresolved issue in PPO**.
> > >
> > > We briefly review subsequent works that attempt to improve PPO-Clip, highlighting their heuristic nature:
> > > - **DAPO** (NeurIPS '25, 1600+citations) [1] is the most popular variant of GRPO, which proposed Clip-Higher, tuning the hyper-parameter of PPO-Clip up-boundary from 0.2 to 0.28.
> > > - **GMPO** (ICLR '26) [2] and **GSPO** (382 citations) [3] are the previous-SOTA RL algorithms, which proposed to clip on the sequential product of importance ratios.
> > >
> > > Other high-impact LLM RL algorithms are also largely heuristic:
> > > - **ORZ** (NeurIPS '25) [4] tuned the hyper-parameter $\lambda$ of PPO GAE from 0.95 to 1.
> > > - **CPPO** (NeurIPS '25) [5] proposed dropping the trajectories with low advantages.
> > > - **VinePPO** (ICML'25) [6] proposed to re-sampling from certain middle states.
> > > - **DrGRPO** (COLM'25) [7] proposed removing the standard deviation in GRPO advantage estimation.
> > > - **GPG** (NeurIPS'25) [8] proposed to replace the standard deviation of GRPO advantage estimation with optional normalization techniques.
> > >
> > > **RIPO distinguishes itself by providing fundamental theoretical insights that transcend these empirical modifications.**
> > > - We theoretically identify that exploration collapse stems from the geometric mismatch between the Euclidean metric used in PPO-Clip and the intrinsic Riemannian manifold of policies, which is a **decade-old, unresolved issue** since the inception of PPO.
> > > - We propose the mathematically grounded algorithm RIPO to rectify this geometric flaw.
> > > - Unlike previous works, we provide a theoretical analysis of how RIPO balances the bias-variance trade-off, a fundamental challenge in policy optimization that heuristic methods fail to address.
> > >
> > >
> > >
> > > ### **2.SOTA Performance and Universal Effectiveness with Broader Evaluation**
> > >
> > > While many recent top-tier RL works for LLM Reasoning (e.g., GRPO [9] in Nature, DAPO [2], ORZ [3], CPPO [4] in NeurIPS '25, GMPO [5] in ICLR '26 and VinePPO [6] in ICML'25) **only focus on Math**, RIPO has been validated across three distinct domains: **Math, Coding and Search**.
> > >
> > > Moreover, we evaluate across 15 distinct benchmarks (seven for Math, four for Coding, and four for Search), **exceeding the current community standard for empirical proof in algorithmic contributions**.
> > >
> > > RIPO is currently the **SOTA RL algorithm for LLM reasoning**, outperforming GRPO by an average of 30% in Math and 15% in Coding/Search. We believe these results persuasively validate the **broad impact** of RIPO.
> > >
> > > For your extra questions about traditional RL training, we trained in environment of (NoFrameskip-)v4, with 20M steps for Atari and 1M steps for MuJoCo, and seed=42 to ensure convergence and strict reproducibility.
> > >
> > > ---
> > >
> > > We truly appreciate the time and effort you have invested in reviewing our manuscript. Given this detailed comparison with current literature, we believe the significance and impact of RIPO, as a theoretically grounded and practically superior algorithm, are now firmly established. **We believe RIPO represents a high-impact advancement for LLM RL reasoning and look forward to its contribution to the field.**
> > >
> > > If our clarifications have further increased your confidence in the technical value of our work, we would be most grateful if you would consider reflecting this in your final evaluation.
> > >
> > > ---
> > >
> > > [1] DAPO: An Open-Source LLM Reinforcement Learning System at Scale (NeurIPS 2025)
> > >
> > > [2] Geometric-Mean Policy Optimization (ICLR 2026)
> > >
> > > [3] Group sequence policy optimization (cited by 382 times)
> > >
> > > [4] Open-Reasoner-Zero: An Open Source Approach to Scaling Up Reinforcement Learning on the Base Model (NeurIPS 2025)
> > >
> > > [5] CPPO: Accelerating the Training of Group Relative Policy Optimization-Based Reasoning Models (NeurIPS 2025)
> > >
> > > [6] VinePPO: Refining Credit Assignment in RL Training of LLMs (ICML 2025)
> > >
> > > [7] Understanding R1-Zero-Like Training: A Critical Perspective (COLM 2025)
> > >
> > > [8] GPG: A Simple and Strong Reinforcement Learning Baseline for Model Reasoning (NeurIPS 2025)
> > >
> > > [9] DeepSeek-R1 incentivizes reasoning in LLMs through reinforcement learning (Nature 2025)

---

### Decision · Program_Chairs · 2026-04-30

**Decision:**

Accept (regular)

**Comment:**

I recommend acceptance. The reviewers were unanimous in recommending acceptance.

The paper provides an interesting new perspective on clipping in PPO and proposes an alternative. There is a derivation of a new update via taylor approximations and fairly comprehensive experiments showing consistent gains.

One caveat is that I think the authors could tone down the rhetoric in places. The new approach seems to be a nice advancement, but it does not give any guarantees about exploration and the paper does not really prove any fully worked proof of collapse in PPO (just some sketched out paragraphs of examples).